# Volatile Organic Compounds Released by *Oxyrrhis marina* Grazing on *Isochrysis galbana*

Charel Wohl *, Queralt Güell-Bujons, Yaiza M. Castillo, Albert Calbet and Rafel Simó *

Department of Marine Biology and Oceanography, Institut de Ciències del Mar, ICM-CSIC, 08003 Barcelona, Catalonia, Spain; queraltguell@icm.csic.es (Q.G.-B.); yaiza@icm.csic.es (Y.M.C.); acalbet@icm.csic.es (A.C.)
* Correspondence: cwohl@icm.csic.es (C.W.); rsimo@icm.csic.es (R.S.)

**Abstract:** A range of volatile organic compounds (VOCs) have been found to be released during zooplankton grazing on microalgae cultivated for commercial purposes. However, production of grazing-derived VOCs from environmentally relevant species and their potential contribution to oceanic emissions to the atmosphere remains largely unexplored. Here, we aimed to qualitatively explore the suite of VOCs produced due to grazing using laboratory cultures of the marine microalga *Isochrysis galbana* and the herbivorous heterotrophic dinoflagellate *Oxyrrhis marina* with and without antibiotic treatment. The VOCs were measured using a Vocus proton-transfer-reaction time-of-flight mass spectrometer, coupled to a segmented flow coil equilibrator. We found alternative increases of dimethyl sulfide by up to 0.2 nmol dm$^{-3}$ and methanethiol by up to 10 pmol dm$^{-3}$ depending on the presence or absence of bacteria regulated by antibiotic treatment. Additionally, toluene and xylene increased by about 30 pmol dm$^{-3}$ and 10 pmol dm$^{-3}$, respectively during grazing only, supporting a biological source for these compounds. Overall, our results highlight that VOCs beyond dimethyl sulfide are released due to grazing, and prompt further quantification of this source in budgets and process-based understanding of VOC cycling in the surface ocean.

**Keywords:** volatile organic compounds; grazing; Vocus; PTR-MS; *Isochrysis galbana*; *Oxyrrhis marina*

## 1. Introduction

Biological processes in the world's oceans produce a plethora of volatile organic compounds (VOCs) that can be released to the atmosphere [1]. In seawater, VOCs can act as a source of energy and carbon for bacteria [2,3] or as messenger molecules in microbial interactions [4–7].

Many of these compounds are released through the interaction between members of the microbiota. One of these processes is grazing by microzooplankton (herbivore protists) on photosynthetic protists (microalgae) [8]. Additionally, bacterial activity can be a source [9–12] or a sink [8] of VOCs to the water column and strongly influence the volatile profile associated with certain phytoplankton [13].

A range of VOCs have been found to be emitted during grazing on species during cultivation for commercial purposes [14]. One study found that, amongst others, pentane, 2- and 3-methylfuran, 3-methylhexane and 3-pentanone were produced as a consequence of the rotifer *Brachionus plicatilis* grazing on the alga *Microchloropsis salina* [15]. The ecological roles of these compounds remained speculative, but the authors suspected that these compounds act as grazer deterrents or allelochemicals, and are generated as by-products of oxidative stress [15]. Sauer et al. [16] reported that monoterpenes decreased and a compound of likely molecular formula $C_4H_7N$ and $NH_3$ increased temporarily as a grazer of the genus *Tetrahymena* consumed the cyanobacteria *Synechococcus elongatus* in the culture. They hypothesised that the decrease in monoterpenes was due to loss in algal biomass, while the transient peaks in nitrogen-containing compounds could be due to a defence

mechanism [16]. Previous experiments had shown the release of β-cyclocitral and β-ionone upon grazing of a rotifer on the phytoplankton *M. gaditana* [17] and *M. salina* [18]. Recently, β-cyclocitral and β-ionone have also been observed in grazed cyanobacterial cultures and their non-grazed control culture [19]. These two compounds belong to a class of compounds coined apocarotenoids, which originate from the degradation of the photosynthetic pigment β-carotene [20]. The roles of these two compounds in algae is rather unknown, but in terrestrial plants β-cyclocitral fulfils allelopathic roles related to stress signalling [20]. For short periods, β-cyclocitral has been found to increase swimming velocities of *Daphnia magna* [21], illustrating a potential role in chemical signalling.

One thoroughly investigated example of a gas produced from grazing in the ocean is dimethyl sulfide (DMS). Dimethyl sulfide is produced through the action of bacterial or phytoplanktonic lyases on the precursor molecule dimethylsulfoniopropionate (DMSP) [22]. Dimethylsulfoniopropionate is largely produced by phytoplankton, with production rates that vary widely between phytoplankton species. Phytoplankton species can be divided into high and low DMSP producers [23–25]. Beyond grazing, algal DMSP is released into seawater through exudation, physiological stress, cell death by autolysis or viral lysis [26–28]. Dimethylsulfoniopropionate and its lyase degradation products, DMS and acrylate, have been shown to influence microzooplankton grazing. Although contradictory outcomes of the net effect, attraction or repulsion, have been reported [29–31]. Dimethylsulfoniopropionate is also degraded by bacteria into methanethiol [32], which is a much more poorly studied VOC, largely due to the difficulty of measuring dissolved methanethiol in seawater. Our current understanding is that most DMSP follows the degradation pathway into methanethiol while only a smaller fraction is transformed to DMS [33,34]. The production of methanethiol is thought to partially depend on bacterial sulfur demand, where bacteria would preferentially produce DMS if their sulfur demand is satisfied [35]. Methanethiol on the other hand seems to be an intermediate, produced during incorporation of sulfur from DMSP into cellular macromolecules [33]. Sea-to-air fluxes and seawater measurements suggest that methanethiol concentrations are approximately 20% of the DMS concentration [36–38], but this ratio can vary hugely [39]. This is because bacterial consumption is much faster for methanethiol than for DMS [32,40,41]. Beyond methanethiol and DMS, a number of sulfur-containing VOCs have recently been measured from an induced phytoplankton bloom [36].

Evidence exists for the production of benzene, toluene and xylene from axenic phytoplankton cultures and mesocosm studies [42]. This points to a biological source in seawater, supported by field measurements in the ocean [43,44]. Toluene is produced by some naturally occurring bacteria [45], including bacteria isolated from surface seawater in Antarctica [10]. An enzyme has been identified in bacteria that is responsible for toluene production [46]. However, very little is known about the actual oceanic biological production mechanisms as hitherto benzene, toluene and xylene have been labelled as anthropogenic pollutants [47]. Their largest source to the atmosphere is from anthropogenic activity, such as fuel evaporation and combustion, spillage, solvent use, refining of gasoline, landfill wastes and coal-fired stations [48]. Fuel spillage is also a source of benzene, toluene and xylene to the seawater [49]. The largest sink of these simple aromatics in seawater is probably bacterial consumption [50,51].

All in all, the floor was ripe to further explore the suite of VOCs produced from microzooplankton grazing and bacterial activity. In this experiment, we aimed to focus on compounds and phytoplankton species that are relevant for marine biogeochemistry. Thus, we decided to measure the VOCs produced in grazing experiments at high resolution using proton-transfer-reaction time-of-flight mass spectrometry (PTR-ToF-MS) coupled to a segmented flow coil equilibrator (SFCE). We adopted a mass scanning approach for an untargeted exploration of compounds, followed by high-time resolution monitoring of specific ions of target VOCs identified during the exploration. Our objective was to work with a highly simplified monospecific predator grazing on a DMSP-containing prey, making use of the bacteria naturally abundant in the culture. We used the marine het-

erotrophic dinoflagellate *Oxyrrhis marina* as a model grazer, and the haptophyte microalga *Isochrysis galbana* as the prey. In contrast to previous studies, we used an environmentally relevant prey in our experiments, rather than species cultivated for commercial interest. Haptophytes are environmentally relevant as they may account for 30–50% of the total photosynthetic standing stock in the oceans [52]. *Isochrysis galbana* is a well-characterised species known to produce large amounts of DMSP [28] while lacking DMSP lyase to cleave DMSP into DMS [53]. *Oxyrrhis marina* is frequently used as a model grazer [54] because of its voracious feeding which is generally well studied [55]. Although it is not typically found in the open ocean, *O. marina* is widely distributed in coastal habitats of the Northern Hemisphere, south of 63°N [56]. *Isochrysis galbana* and *O. marina* were also chosen because they readily grow up to high cell concentrations under antibiotic treatment. We used antibiotics to elucidate what compounds were produced as a direct consequence of the grazing activity or as products of bacterial processing. Short experimental timescales were implemented to avoid potential bottle effects or lab air to culture VOC exchange and allow for comparison between cultures [56]. By continuously measuring VOCs from a phytoplankton culture as it was grazed down over the course of approximately five hours, we expected to detect the VOCs produced during grazing that could be of most interest for atmospheric chemistry.

## 2. Materials and Methods

### 2.1. Phytoplankton and Microzooplankton Cultures

The microzooplankton strain *O. marina* (ICM-ZOO-OM001) was originally isolated by A. Calbet in 1996 from the NW Mediterranean coast [55]. The prey chosen for growing the culture was *Rhodomonas salina* strain K-0294 in exponential growth phase. *Rhodomonas salina* is a weak DMSP producer [57]. By growing the grazer on a low-DMSP producer, we aim to keep DMSP concentrations in the grazer reasonably low and thus avoid potential interferences with our experiment. The prey used in the grazing experiment was *I. galbana* strain CCMP 1323. The three cultures were cultivated in autoclaved filtered seawater, which was supplemented with f/2 nutrients for the two microalgae. The growing phase prior to the experiment was performed on a 14:10 h light to dark cycle, at 19 ºC and $50 \; \mu mol \; m^{-2} \; s^{-1}$ of light, similar to Li et al. [28]. The cell concentration of the cultures and cell volume were monitored using a MultisizerTM 3 Coulter Counter (Beckman Coulter, Indianapolis, IN, USA). Two batches of the grazer and two of the prey were grown for each culture. One was treated twice with a mix of three antibiotics: kanamycin (Merck K1377) at $1000 \; mg \; dm^{-3}$, neomycin (Merck N6386) at $250 \; mg \; dm^{-3}$, and penicillin G (Merck P3032) at $1000 \; mg \; dm^{-3}$, one time at the start of the culture and again 24–48 h before the experiments. *Oxyrrhis marina* was starved for three days before the grazing experiments, until most of their vacuoles were empty and no prey was found in the culture as verified by microscopy. *Isochrysis galbana* was grown to an exponential growth phase for the grazing experiment.

### 2.2. The Grazing Experiment

Cultures of the grazer, *O. marina*, and the exponentially growing prey, *I. galbana*, were mixed on the day of the experiment in dark glass bottles of $2.5 \; dm^{-3}$. Two mixtures, one with and one without antibiotic treatment, are coined here "grazing cultures". As soon as possible after mixing grazer and prey, grazing cultures with and without antibiotic treatment were measured alternatingly for VOCs over five hours by passing the culture through an SFCE equilibrator coupled to a Vocus PTR-ToF-MS (see Appendix A) [58]. Through frequent, repeat measurements of the same grazing cultures over time we aim to increase statistical significance of our observations. These repeat measurements essentially replace biological replicates, which were not possible to achieve because of workload, large number of cells for VOC analysis and requirement of multiple SFCE-PTR-ToF-MS for true replication. Replicate experiments done one after another would have involved using other batches of cultures.

Additionally, at the beginning of the experiment and hourly thereafter, we measured VOCs in each of the cultures of grazer and prey in isolation as well as in the culture medium alone, kept in 0.5 dm$^{-3}$ dark bottles, to check for possible contamination from lab air. Time zero hereafter is defined as the timepoint at which we began the VOC measurements. The cultures measured during the experiment were all diluted in the same batch of autoclaved filtered seawater carefully syphoned in the sampling glass bottles to avoid atmospheric contamination prior to the experiment. Starting cell concentrations of each component in the individual grazer and prey cultures were chosen to be nearly equal to the cell concentrations of each component in the grazing culture to allow direct comparison of VOC concentrations in the grazing and parent cultures. Starting concentrations of *I. galbana* were around 5–6 × 10$^4$ cells cm$^{-3}$ and starting concentrations of *O. marina* were approximately 3–4 × 10$^3$ cells cm$^{-3}$.

### 2.3. Auxiliary Measurements

During the grazing experiments, we took samples for Coulter Counter and Flow Cytometry to document cell concentration and cell volume at times 0, 15, 30, 60, 90, 120, and 300 min after mixing prey and grazer. Samples for epifluorescence microscopy were also collected at times 0, 60, 120, and 300 min to obtain further evidence of the grazing process and quantify the number of *O. marina* cells that had ingested prey. Microscopy samples were fixed with 10% glutaraldehyde and counterstained with 0.5 mg dm$^{-3}$ DAPI (4′-6-diamidino- 2-phenylindole). Images were automatically acquired using a Zeiss Axio Imager Z2m epifluorescence microscope (Carl Zeiss, Berlin, Germany) connected to a Zeiss camera (AxioCamHR, Carl Zeiss MicroImaging, S.L., Barcelona, Spain) at 630× magnification through the AxioVision 4.8 software. The DAPI signal was observed using the UV filter set (370/40 nm excitation, 425/46 emission, and FT 395 beam splitter), while the cell's chlorophyll was observed using a filter set specific for chlorophyll (585/35 nm excitation, 615 LP emission, and FT 570 beam splitter). All pictures were taken using the same intensities and exposure times (15 ms for DAPI and 10 ms for chlorophyll). Nutrient concentrations (total nitrate plus nitrite, ammonia, silicate, and phosphate) were measured at time zero and after five hours in the parent cultures. Nutrients in the grazing cultures were additionally measured after 120 min.

### 2.4. Measurement of Dissolved VOCs

A commercially available Vocus PTR-ToF-MS (PTR-TOF-MS; TOFWERK AG, Thun, Switzerland, Vocus Scout) [58] was coupled to an SFCE equilibrator [59]. Details on the Vocus operation and settings, humidity considerations and data processing are provided in Appendix A.

The inlet of the SFCE equilibrator consisted of an 80 cm PFA tube (outer diameter 6.35 mm, wall thickness, 1.19 mm), which was used to draw samples from the bottom of glass bottles containing the cultures—exactly the same as discrete VOC sampling with the SFCE. We manually rapidly switched between measurement of the different cultures by moving the sampling tube from one bottle to another. Each culture or culture medium was measured for about 3 min which gave a stable signal of 2 min. Between samples, we briefly measured Milli-Q water to distinguish samples during data post-processing. Cultures were gently swirled approximately every 15 min to avoid sedimentation of the microorganisms.

The SFCE equilibrator was reproduced as before [59] with a few noteworthy modifications: (a) a 15 m long segmented flow tube was used to achieve a high degree of equilibration of less soluble compounds (e.g., isoprene); (b) the zero air carrier gas was supplied from a zero air generator (Vocus PTR Clean Air System: ZeroAir, TOFWERK AG, Thun, Switzerland) and the flow was controlled by daily flow-checked needle valves; and (c) the water flow into the equilibrator was reduced to 20.5 cm$^3$ min$^{-1}$ to reduce the culture volume requirement for this experiment. This resulted in operating the water flow-controlling peristaltic pump at 27 rounds per minute which should have been gentle enough to avoid damage to the microorganisms and subsequent artefactual gas release.

The equilibrator headspace air was transported to the main inlet of the Vocus via a 1.2 m PFA tube (outer diameter 3.18 mm, wall thickness, 0.77 mm) equipped with a vent to the atmosphere for excess air flow.

Equilibrator headspace mole fractions were calculated from a multi-compound gas standard. From that, dissolved concentrations were calculated assuming full equilibration of gases more soluble in water than benzene and toluene and accounting for the reduced water flow rate in the purging factor. More details are provided in the Appendix A.5.

To test for significance, we use the repeated measures ANOVA test [60] applying it to pairwise comparisons of time series. We chose this statistical test because it allows to test for significant differences of the same repeatedly measured sample, thus accounting for repeat measurements not being statistically independent. To account for differences in time resolution, which are incompatible with the ANOVA test, higher resolution time series were subsampled to the nearest located measurement of the lower resolution time series. In the text, we state the *p*-value to give the significance level and chose a significance threshold of 0.1 to account for the high dynamicity of the study system.

## 3. Results

### 3.1. Biological Results for Culture Grazing

*Isochrysis galbana* cell concentrations decreased rapidly in both the antibiotically and non-antibiotically treated grazing culture over the first 90 min (Figure 1A). The decrease was slower afterwards (Figure 1A) (O_I_B culture, *I. galbana* cell concentrations vs time: linear regression slope $\pm$ 95% confidence interval; 0 to 90 min $-371 \pm 285$, 90 to 300 min $-65 \pm 59$). The same pattern was reflected in the ingestion rates (Figure 1B), with the particularity that they were on average 21% lower without antibiotics (O_I_B culture, ingestion rates vs. time: linear regression slope $\pm$ 95% confidence interval; 0 to 90 min $-0.036 \pm 0.036$, 90 to 300 min $-0.006 \pm 0.030$). Coulter Counter measurements confirmed that the cell concentrations of *O. marina* (Figure 1A) did not increase substantially during the five-hour experiment. The average cell volume of *O. marina* rapidly increased during the first 90 min and at a slower rate thereafter (Figure 1C). Microscopy images of *I. galbana* cells inside *O. marina* were further used to confirm that the *O. marina* cells were actively ingesting the selected prey during our experiment (Appendix B). These fluorescence microscopy images showed that a single cell of *O. marina* had ingested multiple *I. galbana* cells. They also showed that at time zero, already 66% of the *O. marina* cells had prey inside i.e., monitoring of the experiment started with the grazing already in progress. By 120 min, 95% of the *O. marina* cells contained prey as determined by microscopy. Overall, these observations suggest that there was very intense grazing from the onset of the experiment until approximately 90 min. Then onwards, grazing was slower, while *O. marina* cells did not start to divide yet.

Nutrient measurements (Appendix C) confirmed that over the course of the experiment, phytoplankton did not deplete the nutrients supplied in the culture media. Ammonium concentrations substantially increased in the grazing culture without antibiotic treatment (about 2-fold) and in the *I. galbana* culture without antibiotic treatment (about 70-fold). This is likely because bacteria present in the cultures produced ammonia from nitrogen-containing organic matter released during prey growth and grazing-induced prey cell lysis [61,62]. In contrast, ammonium concentrations in antibiotically treated cultures of *I. galbana* and *O. marina* remained essentially constant over the course of the experiment. Only a very small increase in ammonia concentration (20%) was observed in the grazing culture treated with antibiotics.

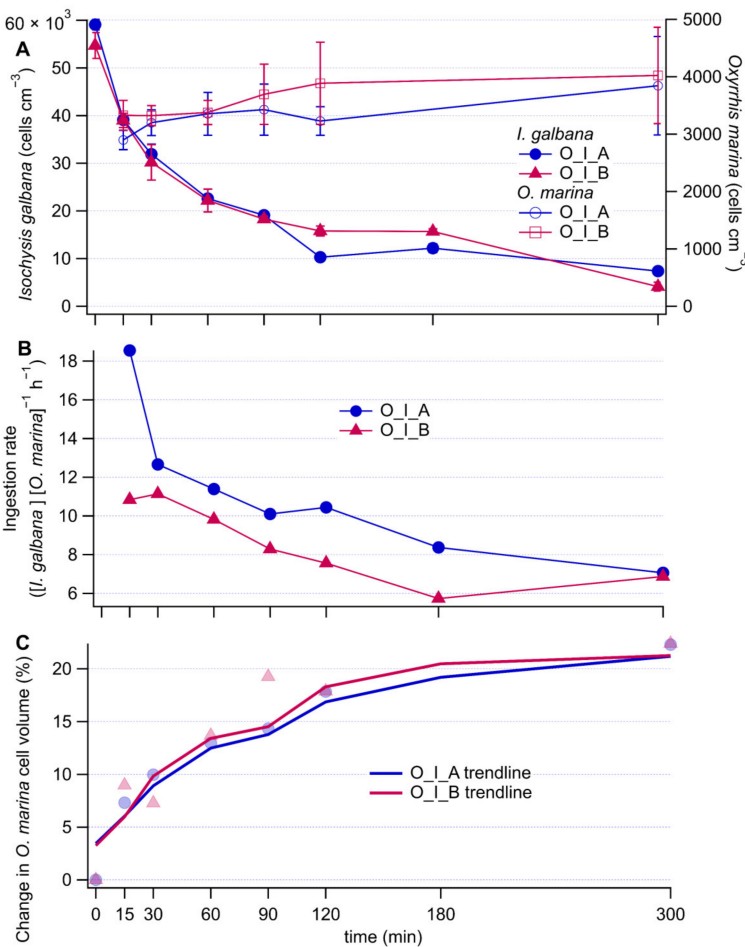

**Figure 1.** Time series of *I. galbana* (prey) and *O. marina* (grazer) concentrations (**A**), ingestion rates (**B**), and percent increase of *O. marina* cell volume (**C**) relative to the initial time point. In the legends, O stands for *O. marina*, I for *I. galbana*, A for antibiotic treatment and B for no antibiotic treatment. Error bars represent measurement noise from repeat measurements of the same sample.

Microscopy results and flow cytometry indicate that bacterial cell concentrations were on the order of $10^6$ to $10^4$ cells cm$^{-3}$ in all cultures, even in those with antibiotic treatment. However, aliquots of the antibiotically-treated cultures plated on agar plates did not show bacterial growth. Similarly, the changes in ammonium concentrations suggest that the bacteria present in the antibiotically-treated cultures were largely inactive. Some of the bacteria observed in the cultures by microscopy and flow cytometry could be antibiotically resistant or inactive bacterial cells or debris. It seems likely that microscopy and flow cytometry count inactive or dead bacteria, which leads to overestimates of the living bacterial cell counts.

### 3.2. Mass Spectral Characteristics to Identify Organic Compounds from Grazing

Figure 2 shows a mass spectrum of the grazing culture without antibiotics. This mass spectrum was calculated as the difference between the spectrum at the end of the experimental period (t = 300 min) minus that at the beginning (t = 0 min). The data are presented as the percentage of the total number of ion-impacts measured by the Vocus detector per second. The numbers do not add up to one hundred in the figure, because prominent peaks related to instrument operation have been set to zero to facilitate interpretation. In the case of measuring the equilibrator headspace, the magnitude of the signal of each individual compound is mostly influenced by the compound's solubility, where more volatile compounds display a higher signal. Thus, this mass spectrum is a reasonable way to explore the more volatile compounds produced in the grazing culture with bacteria.

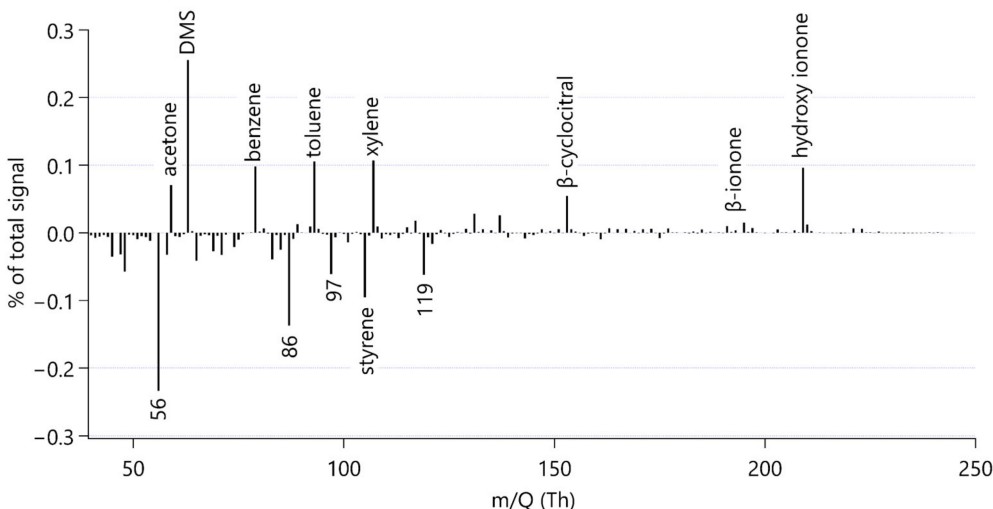

**Figure 2.** Mass spectrum calculated as the difference between the end of the experimental period (t = 300 min) minus the beginning (t = 0 min) of the grazing culture without antibiotic treatment (O_I_B). Identified and discussed peaks are labelled with the compound name or nominal mass. To simplify the interpretation of the spectrum, prominent peaks due to water clusters and some of their isotopes have been set to zero in this graph, namely nominal masses $m/Q$ 55, 57, 73 and 91. Similarly, a prominent peak due to the artefactual acetone-water cluster at $m/Q$ 77 and the instrument parameter $NO^+$ at $m/Q$ 46 are set to zero.

Each bar in Figure 2 represents a different compound measured by the Vocus. This illustrates that the instrument can detect many VOCs in the headspace of a phytoplankton culture, some of which are of very low intensity and not discussed here.

Figure 2 is useful to visually scan for VOCs emitted or consumed during grazing, where potential production is indicated by a prominent positive bar. In our time series analysis, we will focus on compounds which; (a) increased during the experiment; (b) can be identified reliably; and (c) were detected at relatively high intensity.

Notable increases in ion intensity were observed for ions at nominal mass $m/Q$ 59, 63, 79, 93 and 107. By matching high resolution peaks with expected masses based on elemental formula, these peaks were attributed to acetone, dimethyl sulfide, benzene, toluene and the sum of xylenes respectively, in line with common mass assignments [63]. We did not observe other peaks at these nominal masses. The instrument cannot distinguish isomers and thus we cannot tell what isomer of xylene (*o*-, *m*- or *p*-) or ethylbenzene is present in the sample. Hence, we are reporting this signal as "Xylenes" (see Appendix A).

By comparing the time series of the grazing culture with those of the culture media alone, we concluded that the increase in acetone and benzene during our experiment was probably due to a flux of these gases from lab air into the culture. Mole fractions of these two gases can be very high in indoor air [64] and we do not expect any changes in concentration in the culture media alone. Consequently, our results do not appear to support a grazing source for acetone and benzene. Similarly, we could not observe an increase in the acetaldehyde signal due to grazing. Future experiments measuring these gases produced from lab cultures should also focus on rigorously avoiding or quantifying lab air to culture exchange to avoid false positive reporting. For example, Reese et al. [12] had to exclude many peaks from their analysis that were associated with the culture media and the sampling device. Furthermore, Rocco et al. [42] subtracted a background of benzene, toluene and xylene from their culture media.

We additionally recognised an isolated peak at $m/Q$ 49.011, which we identified as methanethiol following the mass assignments of Pagonis et al. [63] and Kilgour et al. [36]. The identity of this peak was additionally confirmed by comparing the peak observed in the experiment and from evasion calibrations of methanethiol by dissolving sodium methanethiolate in seawater [65].

Three ions of *m/Q* 153.107, *m/Q* 193.158 and *m/Q* 209.165 increased in intensity during the experiment. We assigned the molecular formulae $C_{10}H_{16}OH^+$, $C_{13}H_{20}OH^+$ and $C_{13}H_{20}O_2H^+$ to the three ions, respectively, using Tofware software package (ver. 3.2.5, www.tofwerk.com/tofware) developed by Tofwerk and Aerodyne Research Inc. (www.aerodyne.com). This software suggests the most likely elemental formula of a peak given its high-resolution mass. Based on that, we suspect that these ions correspond to α- or β-cyclocitral (CAS number 432-25-7), α- or β-ionone (CAS number 79-77-6) and hydroxy ionone. Note that the Vocus PTR-MS cannot distinguish between isomers of these compounds. These compounds have been identified as common products of carotene degradation in plants [66] and also in algae [5,13,13,18]. Other researchers using PTR-MS alongside GC-MS for identification have identified these peaks as β-cyclocitral, β-ionone and hydroxy ionone [66,67]. In agreement with observations by García-Plazaola et al. [66], we removed influences from the peaks $C_8H_8O_3H^+$, $C_{12}H_8H^+$ and $C_9H_{16}N_2H^+$ from the β-cyclocitral peak using the Tofware software.

By comparing the time courses of cyclocitral, ionone and hydroxy ionone ions in the different cultures, we could not identify a production mechanism as they increased in all cultures over the course of the experiment, but not in the culture media (Appendix D). As there was no difference in concentration between the cultures, a production mechanism could not be inferred. Therefore, although we can confirm the release of α- or β-cyclocitral (CAS number 432-25-7), α- or β-ionone (CAS number 79-77-6) and hydroxy ionone from these microorganism strains (see Appendix D), our results do not allow a more in-depth discussion.

Figure 2 also shows that some ion intensities decreased during the experiment. This could be because bacteria are a sink for these VOCs [68,69], or these compounds may be lost/outgassed to the atmosphere during sampling. We observed notable net decreases in compounds at nominal masses *m/Q* 56, 86, 97, 105 and 119, which could not be reliably assigned. Mass 86 could potentially be 3-pentanone, which has previously been observed to be emitted from the microzooplankter *Brachionus plicatilis* grazing on the phytoplankter *Microchloropsis salina* [15]. For the rest of this analysis, we will focus on compounds with a net production during grazing, but it is important to keep in mind that microorganisms are not only sources but also sinks of VOCs.

### 3.3. Time Series Analysis

In the grazing experiment without antibiotics, the DMS concentrations increased rapidly by 0.08 nmol dm$^{-3}$ during the first 40 min, coinciding with the fastest consumption of the prey, and continued increasing at a slower, constant pace thereafter attaining a total increase in concentration of 0.2 nmol dm$^{-3}$ (Figure 3A). The culture of *I. galbana* alone without antibiotics also displayed an increase of DMS concentration of 0.17 nmol dm$^{-3}$, yet significantly less than the grazing culture (repeated measures ANOVA comparing grazing culture without antibiotics and *I. galbana* culture without antibiotics, $p = 0.005$). Conversely, the antibiotic treatment of both the grazing culture and *I. galbana* culture alone resulted in no substantial increase in DMS concentration. However, antibiotic treatment only resulted in a significant difference in the case of the grazing culture (repeated measures ANOVA, comparing antibiotically treated and non-antibiotically treated; grazing culture, $p = 0.06$, *I. galbana* culture $p = 0.175$). No increase was observed in the *O. marina* cultures, even though a slight decrease occurred in the presence of antibiotics, which resulted in a significant difference between the two *O. marina* cultures (repeated measures ANOVA, comparing antibiotically treated and non-antibiotically treated; *O. marina* culture $p = 0.07$).

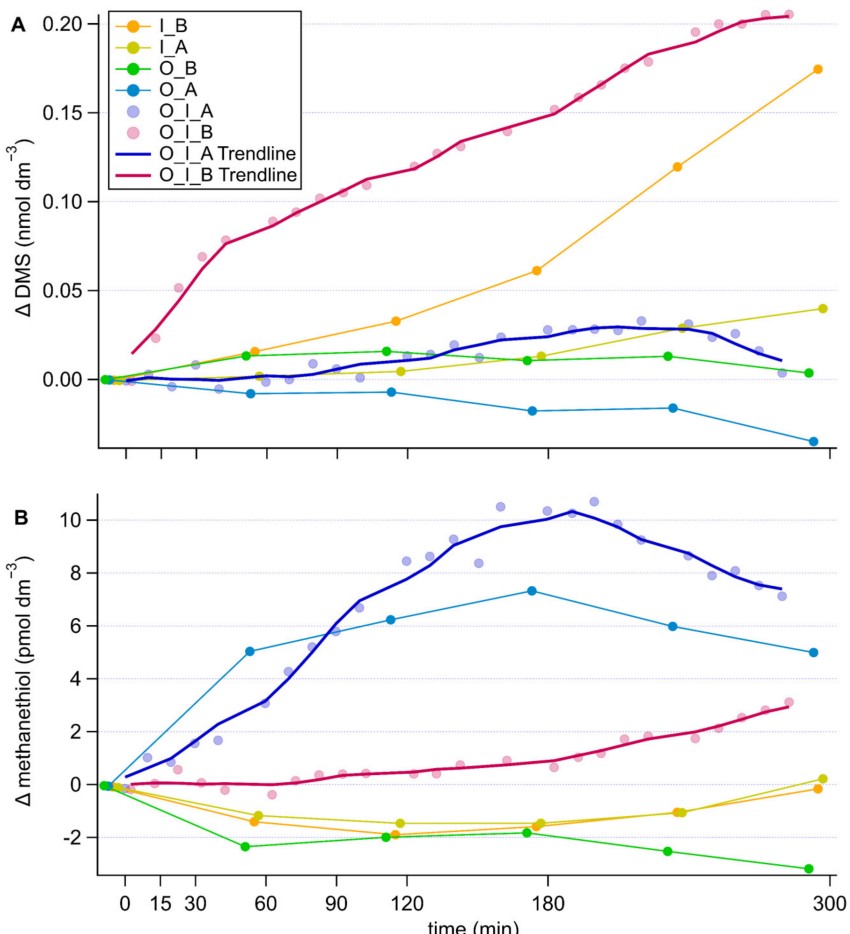

**Figure 3.** Timeseries of changes in DMS (**A**) and methanethiol (**B**) concentration in the different cultures, corrected for the culture medium blank. Components of the cultures are abbreviated as follows, O: *O. marina*, I: *I. galbana*, A: antibiotic treatment, B: no antibiotic treatment. In the grazing cultures, with more frequent data points, the 5-point running average trendlines are shown to help visually tease out trends.

The patterns of methanethiol were somewhat opposite to those of DMS (Figure 3B). In the grazing culture without antibiotics, methanethiol concentrations remained constant during the first 60 min of intense grazing and increased slightly (by 3 pmol dm$^{-3}$) thereafter. In the presence of antibiotics, methanethiol increased immediately from the onset of the measurement period, both in the grazing and the *O. marina* cultures to around 7 pmol dm$^{-3}$ by the end of the experiment. This led to significant differences between the antibiotically treated and non-antibiotically treated cultures containing the grazer (repeated measures ANOVA antibiotically treated and non-antibiotically treated *O. marina* culture, $p = 0.075$, grazing cultures antibiotically treated and non-antibiotically treated $p = 0.094$). Without antibiotics, methanethiol decreased slightly in the *O. marina* culture. No remarkable pattern was found in the *I. galbana* cultures, irrespective of antibiotics.

The aromatic compounds toluene and xylenes showed concentration increases by about 30 pmol dm$^{-3}$ and 10 pmol dm$^{-3}$ respectively only in the grazing cultures and after a lag of approximately 120 min from the onset of the measurement period (Figure 4). This resulted in poor significance levels (repeated measures ANOVA, comparing *I. galbana* culture to grazing culture with and without antibiotic treatment, toluene $p = 0.231$, $p = 0.532$, xylene $p = 0.304$, $p = 0.606$). This time lag was about 30 min longer for toluene in the presence of antibiotics (Figure 4A), an effect that was not recorded for xylenes (Figure 4B). No substantial change in concentration was observed in the cultures of the grazer or the prey alone, irrespective of antibiotic treatment.

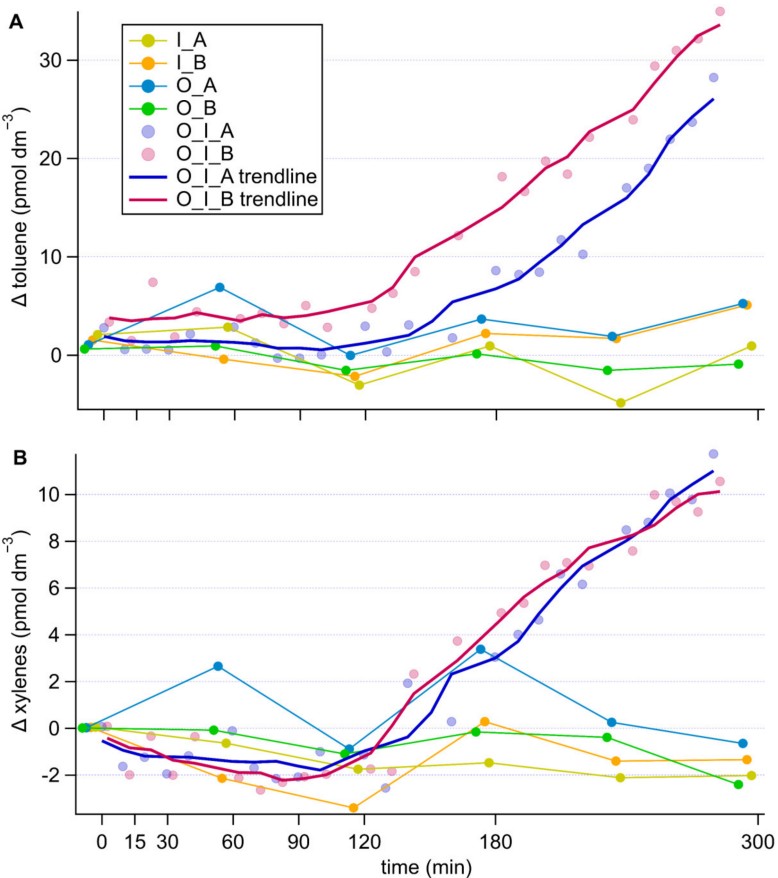

**Figure 4.** Timeseries of changes in toluene (**A**) and xylenes (**B**) concentrations in the different cultures, corrected for the culture medium blank. Components of the cultures are abbreviated as follows, O: *O. marina,* I: *I. galbana,* A: antibiotic treatment, B: no antibiotic treatment. In the grazing cultures, with more frequent data points, the 5-point running average trendlines are shown to help visually tease out trends.

## 4. Discussion

### 4.1. Advantages of Our Experimental Design and Measurement Setup

In most previous studies, VOCs produced by cultures were measured either in the bottle headspace or after bubbling the culture and injecting the sparging gas into a PTR-MS [42,70,71] or a purge and trap gas chromatography/mass spectrometry system [13]. In our experiment, culture volumes were passed through a segmented flow equilibrator, which allowed quick and sensitive measurements. Our setup has the following advantages: (a) we have characterised the degree of equilibration for many compounds and achieved a high level of equilibration for a broad range of VOCs, which results in high sensitivity; (b) the analytical system has a rapid response time of about 30 s; and (c) the segmented flow headspace air is only exposed to very inert PFA, making this setup ideal for the detection of compounds that readily stick to or react with other materials (such as methanethiol, which can be lost to stainless steel [37]); PFA is considered to be the most inert commercially available material for reactive gas measurement [72,73].

With our experimental design, we aimed to distinguish what VOCs were produced due to grazing activity and bacterial processing compared to the grazer and the prey alone. Our use of frequent, rapid response measurements, along with measurements of the grazer and prey alone, allowed us to relate the production of VOCs to the most plausible source using statistics. Furthermore, by measuring the VOCs instantly released from grazing over a relatively short time span, we reduced experimental artefacts, such as lab-air to culture VOC exchange or biological bottle effects.

### 4.2. Grazing Activity

Concentrations of grazer and prey were chosen to lead to near complete consumption of *I. galbana* cells over the course of the experiment. Rapid decreases in cell concentrations of *I. galbana*, especially during the first 90 min (Figure 1A), confirmed the instant response of the grazer to the presence of the prey with high ingestion rates, as expected for a starved *O. marina* culture [55]. A decrease in ingestion rates over the time course of the experiment is due to (a) a reduction in the encounter rates of grazer and prey as prey concentration decreases, and (b) *O. marina* satiation over time [55]. We observed on average 21% lower ingestion rates in the non-antibiotically treated grazing culture. This could be due to (a) small differences in the starting concentrations of *O. marina* affecting competition for prey and (b) *O. marina* probably grazing on bacteria thus alleviating grazing pressure [74]. *Oxyrrhis marina* has been found to feed and grow on bacteria alone [74]. The increase in volume of *O. marina* cells during the experiment (Figure 1C) is related to prey accumulation inside digestive vacuoles. The *O. marina* cell volume increased by 2% more in the grazing cultures without antibiotics, despite lower ingestion rates in this grazing culture (Figure 1B,C), which could be attributed to ingestion of bacteria as well as *I. galbana* [74]. Notably, over the course of the experiment, the grazing cultures contained both intact cells of non-eaten *I. galbana* in exponential growth and grazed cell debris (Figure 1A). *Oxyrrhis marina* cell concentrations did not change substantially during the experiment (Figure 1A), indicating that the grazer cells were mostly actively ingesting and digesting prey and less actively dividing.

### 4.3. Sulfur Compounds

Grazing enhanced DMS production compared to *I. galbana* alone, and DMS concentrations increased faster at the beginning of the grazing experiment, coinciding with faster ingestion rates (Figure 3A). Wolfe et al. [56] also observed rapid production of DMS as *O. marina* was grazing on the high DMSP producer *Emiliania huxleyi*. The very low DMS production in the antibiotically-treated culture confirms that bacteria are key to the production of DMS and that neither *I. galbana* nor *O. marina* possess the relevant enzymes to cleave algal-released DMSP. This is in line with observations by Saló et al. [75] and Niki et al. [53] who found that neither of those organisms possess the ability to cleave DMSP. The DMS increase in the *I. galbana* culture without antibiotics was likely due to bacterial cleavage of the DMSP released by the microalga through exudation or cell death. Li et al. [28] observed an 'acclimatisation period' of *I. galbana* after transfer into culture media resulting in higher DMS concentrations per cell at day 1 compared to day 6 of their experiment. A similar process likely occurred at the onset of our *I. galbana* cultures, but DMS production was observed only when bacteria were present (no antibiotic treatment).

Relatively large accumulations of methanethiol were observed only in the *O. marina*-containing cultures treated with antibiotics (Figure 3B). There is a strong indication that bacteria were net methanethiol consumers in our experiment and antibiotics efficiently arrested consumption and allowed accumulation. The evidence for this is that methanethiol concentrations increased by only half as much in the grazing culture without antibiotics, and even decreased in the *O. marina* culture without antibiotics. Therefore, in our experiments, bacteria seem to be a net sink of methanethiol, while they are a net source of DMS. This is in good agreement with previous observations of rapid bacterial methanethiol consumption [33], and points to marine bacteria as critical players in the regulation of methanethiol concentrations in the surface ocean, being not only sources, but also sinks of methanethiol.

More surprising is the observation of methanethiol production in the *O. marina* cultures with antibiotics. It is not totally clear why this is the case. Because methanethiol sources in seawater are poorly understood, we cannot thoroughly describe the reasons behind this observation. It could be that the grazer produced methanethiol because it had been grown feeding on a low DMSP producer (*R. salina*). There is reported evidence that *O. marina* incorporates sulfur from DMSP-containing prey [75]. Given that the *O. marina* cells were starved before the experiment, it is possible that they kept producing methanethiol as a transient metabolite to sulfur assimilation from DMSP, similar to bacteria [35], and

some was lost through membrane diffusion. The ecological significance of this process remains speculative. *Oxyrrhis marina* grown on a low DMSP producer (*Dunaliella tertiolecta*) has previously been shown to release small amounts of DMS [56], so it seems possible that it could also release methanethiol—another DMSP breakdown product. Overall, the experiment suggests that DMSP degrading bacteria as well as microzooplankton may be biological sources of methanethiol in the pelagic ocean, with bacteria being the main biological sink. This highlights the need for a better process-based understanding of marine methanethiol cycling, especially in light of its non-negligible contribution to the oceanic emission of sulfur to the atmosphere [37].

*4.4. Aromatic Compounds*

The results of the experiment (Figure 4) suggest that toluene and xylenes were released into seawater because of grazing, whereas benzene was not associated with grazing. Additionally, bacteria slightly enhanced toluene production in the grazing cultures. There was a notable lag time to the production of these aromatic compounds compared to that of DMS. The increase in toluene and xylene in the grazing cultures seems to coincide with the increase in methanethiol in the same culture after about 120 min, potentially pointing towards common production mechanism. The observation that the increase in concentration occurred during the second half of the experiment, coinciding with lower ingestion rates, suggests that there are intermediate steps between grazing and toluene/xylenes production. This could indicate that toluene and xylenes are released during digestion of the prey, yet a detailed biochemical mechanism or ecological relevance of this process cannot be speculated at this point. In any case, our experiment provides further evidence for a biological source of toluene and xylenes in seawater, and points to the intervention of grazing and bacteria as a possible explanation for why in situ concentrations of toluene correlate more poorly with chlorophyll *a* than benzene concentrations [44]. Much work remains to be done to elucidate the marine biological sources of these aromatic VOCs that are important for atmospheric chemistry [44].

**5. Conclusions**

Grazing by *O. marina* on *I. galbana* was shown to increase DMS concentration by up to 0.2 nmol dm$^{-3}$ in this culture by release of DMSP from grazing and subsequent bacterial cleavage into DMS. The grazer also increased methanethiol concentration to up to 10 pmol dm$^{-3}$, potentially from a DMSP-containing diet, however, in the presence of bacteria, methanethiol was rapidly consumed thus preventing accumulation. Toluene and xylenes were also produced by grazing, leading to increases to about 30 pmol dm$^{-3}$ and 10 pmol dm$^{-3}$ over the controls. There was a distinct time lag of about 120 min that suggested intermediate steps in their production. Our results prompt to further identify the VOCs produced during grazing on phytoplankton, and to quantify this source in budgets and process-based understanding of VOCs in the surface ocean. Ultimately, this will help to quantify the impact of grazing on ocean emissions of VOCs and atmospheric chemistry.

**Author Contributions:** Conceptualization, C.W., Q.G.-B. and R.S.; data curation, C.W.; funding acquisition, R.S.; investigation, C.W., Q.G.-B., Y.M.C. and A.C.; methodology, C.W., Q.G.-B. and R.S.; project administration, R.S.; resources, R.S.; supervision, R.S.; visualization, C.W.; writing—original draft, C.W.; writing—review and editing, Q.G.-B., Y.M.C., A.C. and R.S. All authors have read and agreed to the published version of the manuscript.

**Funding:** The SUMMIT project is funded by the European Research Council (ERC) under the European Union's Horizon 2020 research and innovation programme (Project ERC-2018-AdG 834162 SUMMIT to RS, Grant Agreement No. 834162). The ICM-CSIC received funding from the Spanish government through the 'Severo Ochoa Centre of Excellence' accreditation (CEX2019-000928-S).

**Institutional Review Board Statement:** Not applicable.

**Informed Consent Statement:** Not applicable.

**Data Availability Statement:** Data used in this paper were acquired during the experiment and are not currently archived in a data repository. Please contact us if you are interested in the data.

**Acknowledgments:** We would like to thank the Simó Lab members for helpful discussions and administrative support. Many thanks to Ana Sotomayor and Claudia Traboni for providing *Rhodomonas salina* for feeding the *O. marina* culture. We are also very thankful to Veronika Pospisilova, Felipe Lopez-Hilfiker and the rest of the team from Tofwerk for excellent instrument support.

**Conflicts of Interest:** The authors declare no conflict of interest.

## Appendix A. Details Regarding the Operation of the PTR-MS and SFCE

*Appendix A.1. Vocus Settings*

The focussing ion-molecule reactor (FIMR) (Vocus Reactor) was operated at 2 mbar. The Vocus front and back voltages were 600 V and 24 V, respectively, giving an axial voltage of 576 V. The quadrupole around the FIMR was set to 500 V and the frequency was $2.22 \times 10^6$ Hz. The temperature in the reactor was 50 °C. Using these measured FIMR settings, we calculated an electric field strength (E/N) of 134 Td (1 Td = $1 \times 10{-17}$ V cm$^2$). However, due to the location of the sensor measuring the FIMR pressure and the application of an RF field, the true electric field strength experienced by the product ions (protonated VOCs) was somewhat different. Hence, it is best practice to estimate the E/N in a Vocus instrument using the fragmentation pattern of α-pinene [58,76]. By comparing published fragmentation patterns to our observed fragmentation ratio, we determined the true electric field strength to be $80 \pm 5$ Td. Therefore, the electric field applied during this experiment was very low, allowing for relatively soft ionisation and less fragmentation. To keep the total ion current measured by the detector below $10^7$ ions per second and thus prolong detector lifetime, the BSQ voltage was set to 275 V. This changes the flight path of ions of low molecular weight and makes that only a small fraction of highly abundant hydronium ions, lower mass hydronium ion water clusters and their isotopes were transmitted to the detector. At the same time, this reduced transmission of all ions of molecular weight of less than m/Q 59 as determined by transmission curves generated from a multi-compound gas standard. This reduced transmission was accounted for in the calculation of the methanethiol (m/Q 49) concentrations. During the experiment, the Vocus was operated to scan masses up to m/Q 250.

*Appendix A.2. Vocus Calibrations*

The Vocus was calibrated on the day of the experiment using a multi-compound gas standard (nominal concentration 500 nmol mol$^{-1}$; methanol, acetaldehyde, acetone, isoprene, DMS, benzene, toluene, m-xylene, α-pinene in nitrogen, Apel-Riemer Environmental, Inc., Miami, FL, USA). The gas standard was serially diluted in zero air from a Vocus zero air unit (Vocus PTR Clean Air System 2020: ZeroAir, Tofwerk, Thun, Switzerland) using two build-in mass flow controllers, which injected the calibrant gas through a 16th inch tube at the Vocus inlet. At the cited FIMR settings, we assessed the contribution of fragmenting toluene to the benzene signal to be less than 2% using liquid standards measured in the SFCE equilibrator.

*Appendix A.3. Vocus Humidity Considerations*

Equilibrator headspace air was laden with humidity at around 20 °C. Previous generations of PTR-MS instruments used to display a dependence of the signal to sample humidity [77] which had to be considered when coupling to the SFCE equilibrator [59]. Compared to previous PTR-MS instruments, the Vocus uses a very high flow of water vapor from a water reservoir into the H$_3$O$^+$ ion source (20 sccm compared to 1–2 sccm on previous generations). This results in an extremely high amount of hydronium ions, their clusters and water vapor concentration in the FIMR. Krechmer et al. [58] showed that the Vocus does not display a humidity dependence of the signal as the flow of sample air into the FIMR does not substantially alter the humidity therein. Experimentally, we were able to confirm that our instrument does not display a humidity dependence of the signal for the compounds contained in our gas standard and at the humidities

expected in equilibrator headspace air. We note that Novak et al. [37] suggested a humidity dependence of methanethiol in their Vocus, which was probably due to a metal tee piece used in their atmospheric sampling line. Our calibrations with liquid standards of methanethiol did not obviously suggest any such artefact, probably because the SFCE equilibrator does not contain any metal pieces.

*Appendix A.4. Vocus Data Processing*

The Vocus data files were processed using Tofware version 3.2.5, run in Igor 9.0.1.2. To do the high-resolution peak fitting, we used the PTR library published by Pagonis et al. [63] and integrated it into our Tofware peak list using only ions identified by PTR instruments listed by Pagonis et al. [63]. If there was a peak we could not identify using the Pagonis et al. [63] library, we used the library from Yáñez-Serrano et al. [78] or compounds listed in García-Plazaola et al. [66]. This peak list was used to de-isotope the mass spectrum. Details on the peak identification for each individual peak are further provided in the results section.

The xylene signal is reported here as the sum of xylene isomers (*o*-, *m*-, *p*-Xylene and ethylbenzene) based on the following reasoning: The Vocus and SFCE sensitivity depends on the hydronium reaction rate constant and the compound's solubility. The hydronium reaction rate constant of the Xylene isomers ranges between $(2.26–2.32)\ 10^{-9}\ \mathrm{cm^3\ s^{-1}}$ (range: 2% of the average rate of the four isomers) [79], while the solubility of these compounds is poorly constrained and the range of estimates overlaps. Hence we use the Vocus sensitivity and water solubility of *m*-xylene [80] (solubility by Karl et al. [81], temperature dependence by Staudinger and Roberts [82]) to calculate the dissolved concentration in seawater. To calculate dissolved methanethiol concentrations, we used the solubility and temperature dependence recommended by Burkholder et al. [83].

The limit of detection was calculated as three times the measurement noise. Measurement noise, in turn, was calculated as the standard deviation of repeated zero air measurements, as recommended by the Vocus PTR-ToF-MS manufacturer. This gave the following rounded results, presented as measurement noise followed by the limit of detection separated by a comma; DMS 0.0001, 0.0004 $\mathrm{nmol\ dm^{-3}}$, methanethiol 0.05, 0.18 $\mathrm{pmol\ dm^{-3}}$, toluene 0.12, 0.41 $\mathrm{pmol\ dm^{-3}}$, xylene 0.05, 0.17 $\mathrm{pmol\ dm^{-3}}$. Small differences between three times the listed measurement noise and the limit of detection are due to rounding numbers.

*Appendix A.5. SFCE Calibrations*

The SFCE equilibrator was calibrated frequently for a period leading up to the experiment day using liquid standards for evasion calibrations. This was used to confirm consistent near-complete equilibration in the segmented flow tube. The advantage of a high degree of equilibration compared to partial equilibration is that the degree of equilibration is more robust and does not have to be monitored continuously for example by using an internal standard. The following compounds were calibrated using liquid standards; DMS, methanethiol, isoprene, acetone, acetic acid, dimethyl disulfide and toluene. The compounds contained in the gas standard canister (methanol, acetaldehyde, acetone, isoprene, DMS, benzene, toluene, m-xylene, $\alpha$-pinene) were additionally calibrated for by invasion calibrations. These compounds were chosen due to their relatively low toxicity, ready availability, and range in solubility/volatility. Low solubility is defined here as a low value of the Henry's dimensionless water-over-air solubility. Of these compounds, isoprene has the lowest solubility.

By demonstrating near-full equilibration of isoprene in the SFCE equilibrator using the longer equilibration tube, we were able to confirm that compounds of higher solubility also fully equilibrate, due to the effect of solubility on the air-sea exchange transfer velocity (low solubility compounds take longer to fully equilibrate than high solubility compounds due to lower air-sea transfer velocities [84]). By conducting evasion calibrations for acetic acid, we are able to verify that the Vocus and the SFCE equilibrator are inert enough to detect compounds of extremely low volatility [73].

We note that these calibrations were done at a water flow of 40 cm$^3$ min$^{-1}$, while the experiment was carried out at a water flow of 20.5 cm$^3$ min$^{-1}$. Last minute adaptations related to culture growth required us to use this lower water flow. Given that we have full equilibration of isoprene at 40 cm$^3$ min$^{-1}$, we also expect this to be the case at the lower water flow rate, which allows for a longer equilibration time. The measured dissolved concentrations were finally computed using published solubility values with which we found agreement in our calibrations and accounting for the reduced water flow using the purging factor [59].

For compounds we directly calibrate for (standard gas canister, evasion or invasion i.e., DMS, methanethiol, benzene, toluene and xylene), we expect the uncertainty to be less than 15%.

## Appendix B. Microscopy Images

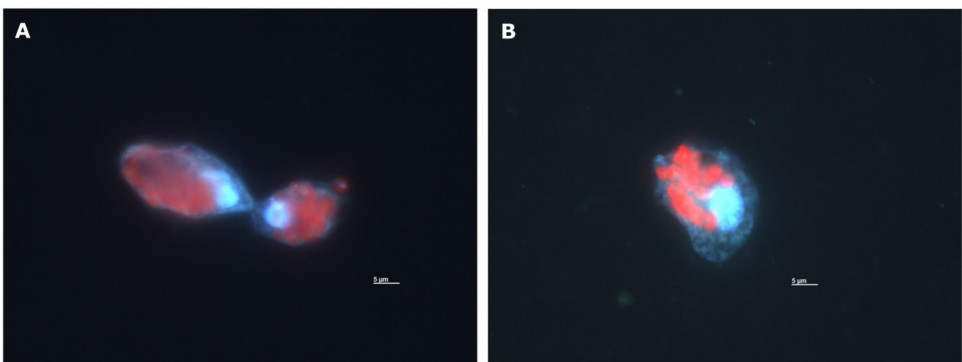

**Figure A1.** Fluorescence microscopy image of *O. marina* cells after grazing on *I. galbana* at the incubation timepoint of 120 min from the antibiotically treated culture (**A**) and the non-antibiotically treated culture (**B**). Blue fluorescence: *O. marina* DNA stained by DAPI (4′,6-diamidino-2-phenylindole). Red fluorescence: Autofluorescence of *I. galbana* chlorophyll *a*. Both images show *O. marina* with several ingested cells of *I. galbana*. Small blue dots in (**B**) are probably bacteria.

## Appendix C. Nutrient Measurements

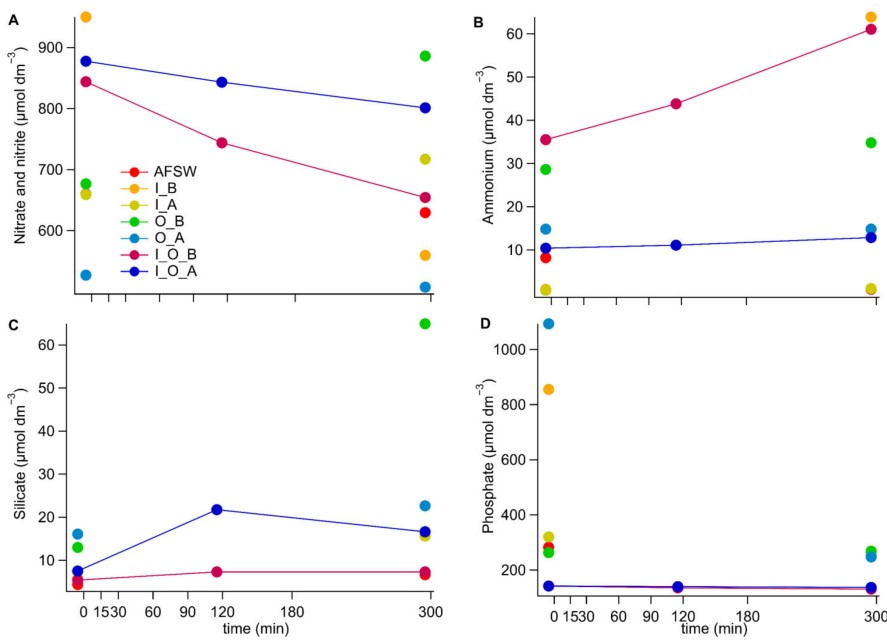

**Figure A2.** Nutrient concentrations of total nitrate and nitrite (**A**), ammonium (**B**), dissolved silicate (**C**) and phosphate (**D**) measured in the different cultures over the course of the grazing experiment. The content of the cultures is abbreviated as follows; O: for *O. marina*, I: for *I. galbana*, A: for antibiotic treatment and B: for no antibiotic treatment, AFSW: autoclaved filtered seawater, which is the culture medium for all cultures.

## Appendix D. Norisoprenoid Time Series

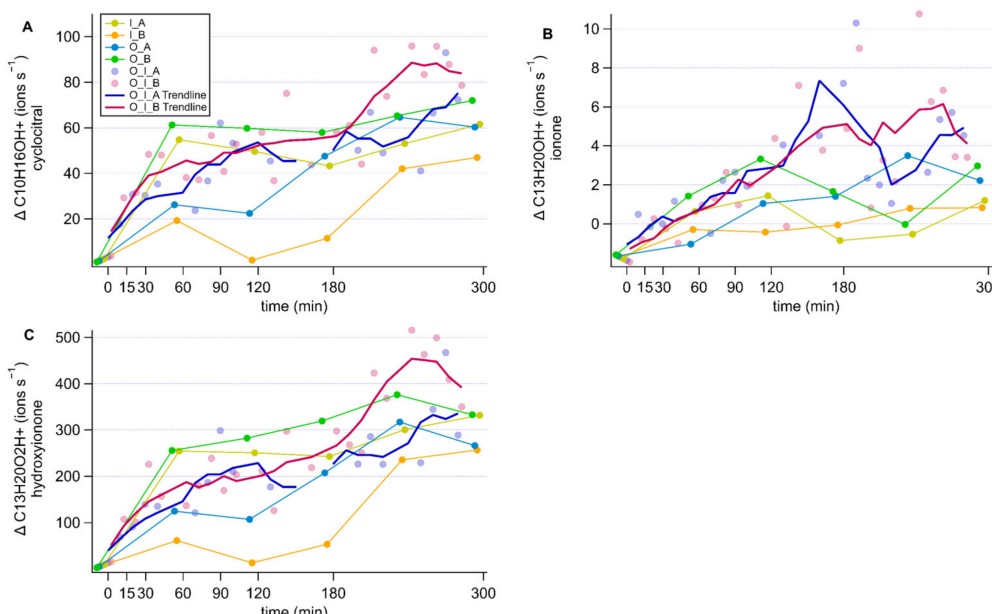

**Figure A3.** Timeseries of the changes in norisoprenoid concentrations expressed in ions s$^{-1}$. Each panel represents a different compound; β-cyclocitral (**A**), β-ionone (**B**) and hydroxy ionone (**C**). Components of the culture are abbreviated as; O: *O. marina,* I: *I. galbana*, A: antibiotic treatment, B: no antibiotic treatment.

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
