# Peer review of "Volatile Organic Compounds Released by Oxyrrhis marina Grazing on Isochrysis galbana"

_2673-1924, doi:10.3390/oceans4020011_

Round 1

Reviewer 1 Report

Comments:

1. page 3 - 2.1 : The prey chosen for growing 102 the culture was Rhodomonas salina strain K-0294 since it is a weak DMSP producer - how long time taken to grow?  the chosen of culture based on what since it it a weak DMSP. 

2. performed on a 14:10 hour light to dark cycle, at 19 ºC and 50 μmol m-2 s-1 of light - based on which reference and justification chosen values.

3. MultisizerTM 3 108 Coulter Counter (Beckman Coulter) - to add city, country.

4. page 3 - 2.2: Vocus PTR-ToF-MS (see below) - to see which section?

5. page 7 - 3.2: We thus refrain from discussing them further but can confirm that our experiments show that these compounds are released from the microorganism strains studied here - unclear statement and need to justify.

6. Page 8- line 325: increased slightly - what is the % different of the slightly increased? 

7.  Page 9 - line 341: was slightly longer - you may write in term of of time for longer. and how long the increment?

8. Page 9-Line 373 - Slightly lower ingestion rates , what is the increment of the ingestion rate as mention in the context?

9. Page 9 - section 4.3: Grazing enhanced DMS production compared to I. galbana alone, and DMS 386 concentrations increased faster at the beginning of the grazing experiment, coinciding 387 with faster grazing rates (Figure 3a) - how do you measured 'increase faster' in the condition? what is the deviation of the increment?

10. How do you measure 'higher initial DMS concentrations'? 

11. Line 399 - increased much less in the grazing culture without ... you may change all word into technical value as this is much clear for reader rather than using word with no meaning such as increased much less: how much? how less? Also check thoroughly through out your contents.

12. More surprising is the observation of methanethiol production in the O. marina culture with antibiotics. It is not totally clear why this is the case, due to a poor  understanding of methanethiol sources in seawater - unclear statement.

Reviewer 2 Report

Wohl et al report the level of VOCs produced due to microzooplankton grazing using laboratory cultures of marine alga and herbivores (heterotrophic dinoflagellates) with/without antibiotic treatment. The subject is interesting to the research community. Their found VOCs beyond DMS are released due to grazing, thus highlighting the ecological significance of VOCs in the surface ocean. The manuscript is generally well-written although some sentences may need improvements. The determination of VOCs is not an easy job, but the authors have made reasonably good measurements. I would suggest more descriptions of the motivations and the details of the results and experimental procedures. Also, some parts of the conclusions may need better justified.

My major concerns are listed below:
1.     I would prefer more explicit research questions to be addressed in the Introduction section with a better connection to the experiments they had performed.
2.     So far, the result section of the paper is not very convincing. It may be improved by including some statistical analysis and proper comparisons. See my specific comments below.
3.     In the discussion sections 4.3 and 4.4, following the reports of individual compounds, we would expect to see a comparison between DMS and other VOCs, and the descriptions of the relevant ecological significances.

Overall, it is a nice work but needs a much deeper explanation of exciting results, and I would recommend it for publication after my above concerns are properly addressed.

Specific comments
1.       L33-44:The authors have listed a number of literature results that organic compounds are released through the interaction between microzooplankton and phytoplankton. I think it would be more logical if they can also provide descriptions of the significance of the relevant compounds. For example, what are the ecological functions and effects of these organic compounds? Please rephrase the sentence accordingly.
2.       L75: Regarding “anthropogenic pollutants” mention here, the authors may need a better explanation of why these compounds belong to anthropogenic pollutants.
3.       L90: Is it common in the estuary? Surprisingly, authors chose a not typically found grazer in the open ocean. This may limit the application prospect of their results when one tries to assess the relevant environmental impacts.
4.       L108: Where are the cell volume data? If the data is not shown, it is recommended to include an explanation here.
5.       L137: What are the actual volumes of samples taken each time? Will the change in the volume in the container affect your results?
6.       L268: “was probably due to a flux of these gases from lab air into the culture”: this conclusion is arbitrary and needs to be justified. More details are needed here to better explain your statements.
7.       L312: The results shown here are very important and should be the central part of the whole paper. However, they are simply described as a trend, much more detail should be added in this paragraph and the paragraph hereafter.
8.       L321: Statistical analysis is required when comparing two sets of data.

Reviewer 3 Report

The work is relevant to the field of reducing volatile organic compounds in the oceans. The scope of the work is adequate, the work consists of all parts necessary for the article.

At work I noticed the following shortcomings that should be corrected:

1. The abstract should be supplemented with the numerical values of the main results obtained during the research;

2. At the end of the introduction, the aim of the work should be presented more clearly;

3. In the methodology, it is necessary to provide information about the measurement limits and measurement errors of VOC measuring instruments;

4. The names of Figures 1 and 3, the arrangement of references (a) and (b) should be given according to the requirements of the journal (see template). It is also recommended to increase the font size of the text in the Figures;

5. The discussion should include a comparison of the numerical values of the results obtained by you with the works of other authors;

6. The conclusions should contain information about the numerical values of the obtained main results (VOC);

The work can be accepted after correcting the mentioned deficiencies.

Reviewer 4 Report

The manuscript has focused on the evaluation VOCs produced due to grazing using laboratory cultures of the marine microalga Isochrysis galbana and the herbivorous heterotrophic dinoflagellate Oxyrrhis marina with and without antibiotic treatment. . The results were reasonable. I would like to recommend the publication of this study if the following improvements could be made.

1)    It is difficult find a novelty in this study. How is it different from previous studies mentioned in the introduction section, and why does author think this study is necessary?

2)    It is mentioned in the text: “Error bars represent measurement noise from repeat measurements of the same sample”, but for a relevant study, all experiments must be performed in triplicate and not just the measurements of the same sample. For relevant conclusion please present error bars from triplicate experiments for all the samples.

3)    The authors mention that: “The increase in volume of O. marina cells during the experiment (Figure 1c) is related to prey accumulation inside digestive vacuoles. The O. marina cell volume  increased slightly more in the grazing cultures without antibiotics, despite lower ingestion  rates in this grazing culture, which could be attributed to ingestion of bacteria as well as. galbana. Notably, over the course of the experiment, the grazing cultures contained both  intact cells of non-eaten I. galbana in exponential growth and grazed cell debris. O. marina celll concentrations did not change substantially during the experiment, indicating that the grazer cells were mostly actively ingesting and digesting prey and less actively dividing” – please support this information with bibliographic sources.

Round 2

Reviewer 1 Report

minor questions from result and discussion section.

Author Response

Response to reviewer 1 comments.

We thank the reviewer for this second round of reviews which pointed out a few minor oversights. 

Point 1: line 437, this has been changed in the manuscript to:

Our use of frequent, rapid response measurements, along with measurements of the grazer and prey alone, allowed us to relate the production of VOCs to the most plausible source using statistics. 

Point 2: line 454, this has been changed in the manuscript to:

The O. marina cell volume increased by 2 % more in the grazing cultures without antibiotics ...

Reviewer 2 Report

I thank the authors for taking all my question points. I have no further comments.

Author Response

We thank the reviewer for their constructive comments.

Reviewer 4 Report

I consider that the manuscript has been improved after my previous recomandation. I kindly recomand the manuscript for publication.

Author Response

(The authors gave the same response as above.)
